# Thermal Stability of Plasma-Sprayed Thick Thermal Barrier Coatings Using Triplex Pro^(TM)-200 Torch

**Shiqian Tao [1,2,3], Jiasheng Yang [2,3,*], Wei Li [1], Fang Shao [2,3], Xinghua Zhong [2,3], Huayu Zhao [2,3], Yin Zhuang [2,3], Jinxing Ni [2,3], Shunyan Tao [2,3,*] and Kai Yang [2,3]**

[1]  School of Materials Science and Engineering, University of Shanghai for Science and Technology, Shanghai 200093, China; 182442562@st.usst.edu.cn (S.T.); liwei176@usst.edu.cn (W.L.)

[2]  The Key Laboratory of Inorganic Coating Materials, Chinese Academy of Sciences, Shanghai 201899, China; shaofang@mail.sic.ac.cn (F.S.); xhzhong@mail.sic.ac.cn (X.Z.); huayuzhao@mail.sic.ac.cn (H.Z.); zhuangyin@mail.sic.ac.cn (Y.Z.); njx@mail.sic.ac.cn (J.N.); kaiyang@mail.sic.ac.cn (K.Y.)

[3]  Shanghai Institute of Ceramics, Chinese Academy of Sciences, Shanghai 201899, China

*  Correspondence: jiashengyang@mail.sic.ac.cn (J.Y.); sytao@mail.sic.ac.cn (S.T.); Tel.: +86-21-6990-6321 (S.T.)

**Abstract:** Segmentation-crack structured yttria-stabilized zirconia (YSZ) thermal barrier coatings (TBCs) were deposited by atmospheric plasma spraying (APS) using a Triplex Pro™-200 gun. In this work, free-standing coating specimens (~700 μm) were isothermally heat-treated in air from 1200 to 1600 °C for 24 h and at 1550 °C for 20 to 100 h, respectively. The thermal aging behaviors such as microstructures, phase compositions, grain growth and mechanical properties were characterized via scanning electron microscopy (SEM), X-ray diffraction (XRD), electron backscatter diffraction (EBSD) and a Vickers hardness test. The results indicated that the as-sprayed coatings mainly consisted of metastable tetragonal (t′-YSZ) phase, but the t′-YSZ gradually partitioned into equilibrium tetragonal (t-YSZ) and cubic (c-YSZ) phases due to yttrium diffusion during thermal exposure, and with an improvement in temperature, the c-YSZ may retain or transform into another yttrium-rich tetragonal (t″-YSZ) phase. The transformation of t-YSZ to monoclinic phase (m-YSZ) has occurred after 1550 °C/40 h heat treatment, and the content of the m-YSZ phase increased with the prolongation of the thermal exposure time. The variations of Vickers hardness have a correlation with pores healing and grain growth, which might be attributed to the coating sintering and m-YSZ phase formation. Furthermore, the growth pattern of the grains was investigated in detail. In service, cracks and pores proceeded along the grain boundaries, especially surrounding the small grains. It is conducive to the engineering application of TBCs fabricated with the Triplex Pro™-200 gun.

**Keywords:** thermal barrier coatings (TBCs); thermal aging treatment; microstructure; grain growth; electron backscatter diffraction (EBSD)

---

## 1. Introduction

Yttria-stabilized zirconia (YSZ) thermal barrier coatings (TBCs) are used to improve gas inlet temperature, thereby improving thermal engine efficiency and prolonging turbine blade life [1,2]. TBCs consist of a layer of top ceramics, which plays a role in reducing the surface temperature of the substrate by preventing heat from flowing to the substrate directly [3]. A typical YSZ coating with low thermal conductivity and favorable thermal shock resistance is usually deposited either by atmospheric plasma spraying (APS) or electron beam–physical vapor deposition (EB–PVD) [4,5]. The thickness of TBCs is generally less than 500 μm. With the increase of the ceramic layer thickness, the thermal insulation performance of TBCs enhances. When the YSZ ceramic layer thickness is increased by 25.4 μm, the surface temperature of turbine parts can be reduced by approximately 4–9 °C [6].

Therefore, increasing the thickness of the ceramic layer is particularly important for developing TBCs with higher thermal insulation performance and longer service life. However, thicker TBCs prepared by the traditional plasma spraying process had large residual stress, low bonding strength and strain tolerance [7–9], so it is difficult to meet the requirements of engineering application. Therefore, only increasing the thickness of the TBCs ceramic layer cannot acquire a coating with excellent performance.

In recent decades, researchers have found that segmentation cracks can release thermal mismatch stress formed across the ceramic layer and improve the strain tolerance of the APS top coatings [10]. In addition, the segmentation crack can effectively restrain the propagation of horizontal cracks in the YSZ coating, and improve the service lifetime and reliability of the TBCs [11]. Many new technologies such as suspension plasma spraying (SPS), solution precursor plasma spraying (SPPS) and electron beam–physical vapor deposition (EB–PVD) were introduced for the coating preparation [12,13]. The EB–PVD coatings with a columnar microstructure usually exhibit typical superior properties like longer thermal cycle lifetimes. However, the EB–PVD process is complicated and deposition efficiency is low. It is much more difficult to prepare thick TBCs than other processes [14,15]. In view of the relatively low cost and practical application of engineering, researchers generally deposit segmentation-crack structured coatings by optimizing the process parameters of atmospheric plasma spraying (APS) such as increasing spraying power, improving substrate temperature, raising feeding rate and shortening spraying distance [16,17]. Considering that, a Triplex Pro™-200 gun has taken into account, the ceramic coating with a better microstructure can be easier achieved with higher spraying power, greater powder feed rate and adjustable spraying distance. The specialty of Triplex Pro™-200 is a three-cathode plasma spraying system and offers distinctly higher particle velocities due to its advanced nozzle design. Because of this capability, the range of reachable porosity can be extended clearly [18]. Richardt et al. [19] investigated the influence of the process parameters on coating characteristics (porosity and hardness). It turned out that both increasing particle velocity and particle temperature cause an increase in coating hardness. The microstructure could be controlled well via changing feedstock powder in the TriplexPro™-200 system. It was noted that the coating with vertical cracks was delaminated in the range of 114–126 thermal cycles, and there was no palpable cracking or delamination at the interface between the bond-coat and top-coat [20]. It revealed that the APS YSZ coatings deposited by the Triplex Pro™-200 torch may have a relatively long service life.

Unfortunately, due to long-term exposure to high-temperature environments, the YSZ coating suffered serious sintering, and subsequently, its microstructure such as porosity, crack states, grain size and phase composition were about to change significantly, which had a notable impact on the mechanical properties of the coating [21]. During thermal treatment, the pores and cracks of the coating healed, and with the extension of thermal aging time and increase of temperature, the coating endured a phase transition from the metastable tetragonal phase to the stable tetragonal phase and cubic phase with yttrium diffusion. The tetragonal phase gradually changed into the monoclinic phase [22]. The formation of the monoclinic phase could lead to a 3–5% expansion of the coating volume [23,24]. The accumulation of the coating compressive stress finally gave rise to the coating peeling off. In addition, the grains increased obviously and the columnar crystals were able to transform into equiaxed crystals [11,25]. As suggested, the greater thermal shock resistance may put down the existence of plenty of columnar grains. Columnar grains in APS YSZ coatings are typically near-perpendicular to the substrate surface and prevent the propagation of cracks along the top-coat/bond-coat interface [26]. Several studies have investigated the thermal stability of high thermal complicant segmented-crack coatings deposited by APS through varying spraying parameters [25]. However, research has been conducted into the effects of thermal aging time and temperature on the thermal stability behavior of APS top coatings produced by the Triplex Pro™-200 gun.

In this study, free-standing segmentation-cracked TBCs were prepared by an atmospheric plasma spraying (APS) system using a Triplex Pro™-200 gun and an optimized APS process accompanied by one much finer feedstock. We aimed to study the effect of thermal aging time and temperature on

microstructure, phase composition, grain growth and mechanical properties of the coatings. It provides some useful guidance for the engineering application of the coating.

## 2. Experimental Procedure

### 2.1. Coating Preparation

The commercially available powder 8YSZ ($ZrO_2$-8 wt.% $Y_2O_3$) was applied to deposit ceramic coatings. The feedstock was Metco 204F (Oerlikon Metco, Switzerland), which was yellow and more refined than traditional powders. The morphology of the YSZ powders is shown in Figure 1. The particles have spherical morphology with the size ranging from 15 to 45 μm and can melt effectively in plasma flow, which was used to acquire coatings with denser microstructure. The as-sprayed 8YSZ TBCs with a thickness of approximately 700 μm were deposited on stainless steel plates by atmospheric plasma spraying (APS) process using the Triplex Pro™-200 torch and Twin-120-A/H-h and Single-120-A/H-h feeders (Sulzer Metco AG, Switzerland) with three hoppers. The spraying parameters are listed in Table 1. Due to the mismatch of thermal expansion coefficients between the substrate and top coating, the free-standing samples functioned as thermal aging treatment were able to detach from the substrates. Ten sets of substrate-free specimens were prepared for thermal aging treatment in a high-temperature chamber furnace (HTK 20/17, ThermConcept, Germany) at 1200, 1300, 1400, 1500 and 1600 °C for 24 h and at 1550 °C for periods of 20, 40, 60, 80 and 100 h, respectively. The samples were put into the furnace and heated at approximately 10°/min to the target temperature. After keeping in an air atmosphere for a selected time, the coatings were cooled in the furnace to room temperature. All the free-standing samples needed to be impregnated with epoxy (conductive resin for EBSD), cut to prepare for the cross-sectional observation and finally polished by routine metallographic methods. Particularly, ion beam polishing was necessary for samples used for EBSD testing.

### 2.2. Coating Characterization

The microstructures of the substrate-free samples were observed by a scanning electron microscopy (SEM, S-4800, HITACHI, Tokyo, Japan) operated in backscattered electron image mode. The information on phase compositions of each specimen before and after heat treatment were characterized by an X-ray diffractometer (XRD, D/max 2550 V, Rigaku Japan) with filtered Cu Kα (40 kV, 40 mA) radiation at a scan rate of 4° $min^{-1}$ in the 2θ range of 10°–90°. Besides, a more detailed scanning at a slower scan rate of 0.2° $min^{-1}$ in the 2θ range of 72°–76°, and the step size is 0.02°. Phase content, distribution and grain growth behavior can be investigated by electron backscatter diffraction (EBSD, Symmetry, Oxford, UK); particularly, the scanning area was approximately 5100 $μm^2$ and the step size was 150 nm. Via EBSD image analysis of a certain coating area, some useful information such as the size, distribution, orientation and grain boundary distribution of the crystal grains and the phase contained in the area can be easily acquired.

The overall hardness of the coating containing these defects can be measured. Vickers hardness measurements were carried out on the cross-section of the coatings after grinding and polishing using a Hardness Tester (Vickers, Tukon-2100B, Instron, Norwood, MA, USA) under the load of 300 g with a dwell time of 10 s. The indentations were applied near the center region on the ten sets before and after thermally treated YSZ coatings. Ten measurement points were taken for each sample and the average results were recorded as the Vickers hardness of the sample. Coating porosity was evaluated by an image analysis method. Generally, vertical cracks that penetrate at least half of the thickness of the coating are called segmented cracks. The ratio of the number of segmented cracks in the entire observation section to the horizontal length of this section is referred to as the segmentation cracks density of the coating. SEM was used to take 10 images with the same magnification as a group and calculate the average value.

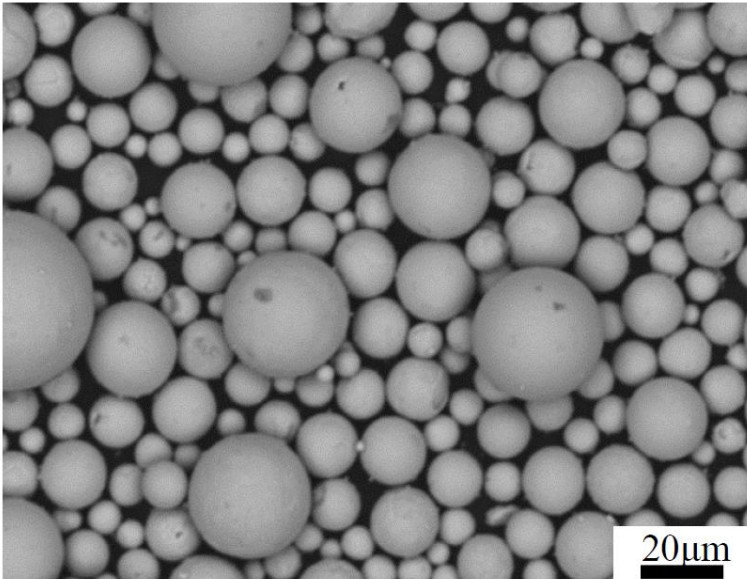

**Figure 1.** SEM images of spherical yttria-stabilized zirconia (YSZ) powders (Metco 204F feedstock) for the ceramic top-coat.

**Table 1.** The spraying parameters applied to the YSZ coating.

| Parameters | Top Coating |
|---|---|
| Plasma torch | Triplex Pro™-200 |
| Plasma gas mixture | He + Ar |
| Plasma gas flow rate (slpm) | He: 4–7 Ar: 40–47 |
| Carrier gas (slpm) | Ar: 2–3 |
| Current (A) | 500–530 |
| Stand-off distance (mm) | 90–110 |
| Powder feed rate (g/min) | 75–105 |

## 3. Results and Discussion

### 3.1. Phase Composition

The XRD patterns of powder and the corresponding coatings before and after thermal exposure at various temperatures for 24 h and at 1550 °C for various times were shown in Figure 2. Compared to the PDF standard card, there was a small amount of monoclinic phase (m-YSZ) in the spraying powder, which may be due to stress accumulation during powder pretreatment. The results indicated that the main phase of the as-deposited coatings was tetragonal, and no peaks corresponding to m-YSZ were observed. Probably because of the rapid solidification and rapid cooling of YSZ splats after impacting the surface of the substrate during the plasma spraying process which limits the diffusion of phases. Figure 2a exhibited the XRD patterns of samples before and after thermal aging at different temperatures for 24 h. It was shown that there was no monoclinic phase existing in the coatings even when the temperature increased to 1600 °C, which denotes that no tetragonal–monoclinic phase transformation occurred in the APS YSZ coatings. From the XRD patterns in Figure 2b, after 40 h thermal exposure at 1550 °C, the m-YSZ peaks at 2θ values of approximately 28.2° (111) and 31.5° (111) emerged, and the m-YSZ peak intensities of the APS coating were significantly higher and narrower as aging time increased. It demonstrated that the phase transition of transformable t-YSZ to m-YSZ occurred as aging time increased and the extent of phase transformation rests with the length of thermal

exposure time. It is reported elsewhere that the phase transition is always accompanied by a volume expansion up to 3–5% expansion, bringing the changes to coating microstructures (crack distribution, crack density and coating porosity), eventually may lead to the thermal barrier coating peeling off.

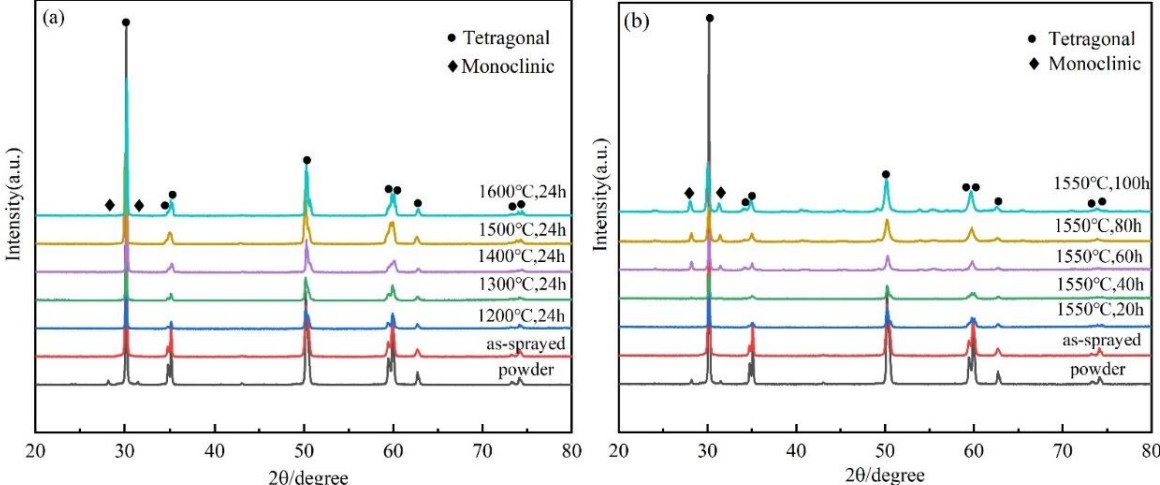

**Figure 2.** XRD patterns of the powder, as-sprayed coating and the coatings after thermal aging treatment: (**a**) top coatings after aging for 24 h at different temperatures (**b**) top coatings after aging at 1550 °C for different times. (Tetragonal is meant as a group of equilibirium tetragonal (t-YSZ), metastable tetragonal (t′-YSZ) and yttrium-rich tetragonal (t″-YSZ) phases).

In order to have a better understanding of the phase transformation phenomenon during thermal exposure, XRD patterns with 2θ between 72° and 76° were given in Figure 3. Results showed that the tetragonal phase of the as-deposited coating is mainly the nontransformable metastable phase (t′-YSZ). There was still a small amount of t-YSZ phase in the as-sprayed coating, which may have come from the original YSZ powders. Upon high-temperature treatment, the t′ phase gradually partitioned into yttrium-deficient t-YSZ and yttrium-rich c-YSZ due to yttrium diffusion, and when the temperature increased up to 1400 °C, the peaks of the c-YSZ disappeared and additional yttrium-rich t″-YSZ peaks formed, which demonstrated that the c-YSZ to t″-YSZ phase transformation took place. For the coatings heat-treated at 1550 °C, t-YSZ peaks disappeared gradually with prolonged aging time, which implied that the coating suffered from the phase transition of t-YSZ into m-YSZ. It is worth noting that the t″ phase is not equivalent to the c phase. The oxygen ions in the t″ phase still occupy a cubic lattice, but the oxygen ions slightly shifted along the c axis of the cubic phase [27,28]. There is no relevant research to confirm that the t″ and t phases have noteworthy impacts on YSZ coating properties. It needs to be pointed out that although the t′-YSZ phase began to decompose at around 1300 °C, it is easier to accelerate the phase transition at higher temperatures. The segmented APS YSZ coatings exhibited similar high-temperature phase stability compared to the conventional APS coating [22,29].

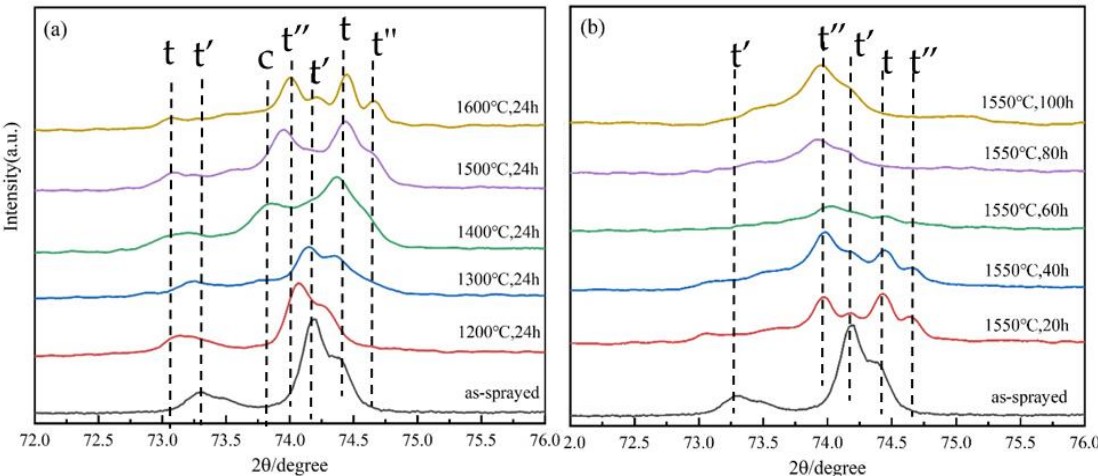

**Figure 3.** XRD patterns of the powder, as-sprayed coating and the coatings after thermal aging treatment with 2θ between 72° and 76°: (**a**) top coatings after aging for 24 h at different temperatures (**b**) top coatings after aging at 1550 °C for different times.

## 3.2. Microstructure

Figure 4 shows the original surface and polished cross-section morphologies of the as-sprayed coatings. It shows that there were few unmelted or semimelted particles in the coating. The powders were well-molten during the spraying process and the change of process parameters had a significant influence on the microstructure of the coating. As seen from Figure 4, there were some obvious vertical cracks and branching cracks in the coating. Currently, vertical cracks that run through at least half the thickness of the coating are usually called segmentation cracks, which can release stress and increase the strain tolerance of the coating under tensile stress. Branching cracks are generally parallel to the coating surface. The branching cracks were connected to the segmentation crack and perpendicular to the propagation direction of the heat flux, which could effectively improve the thermal resistance of the coating. The branching cracks were in favor of increasing the compliance of the coating, but made the coating susceptible to erosion and foreign object damage [30]. The measured segmentation crack density of the coating was approximately 3.5 mm$^{-1}$. The average porosity of the YSZ coating was approximately 2.0% and its structure was much more compact.

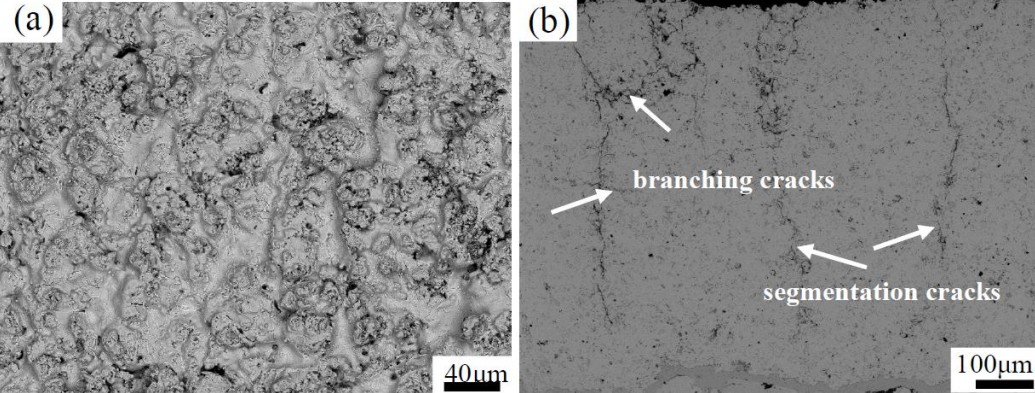

**Figure 4.** Surface morphology (**a**) and polished cross-section morphology (**b**) of as-sprayed APS top coatings.

Figure 5 shows the fracture morphology of as-sprayed coatings. In order to acquire a more detailed and accurate description of the coating microstructure, a higher magnification diagram is shown. As mentioned earlier, the deposition conditions have an important impact on the microstructure of

as-sprayed coatings. The Triplex Pro™-200 torch provided higher power and the powder feed rate was approximately three times higher than the traditional F4MB-XL gun, allowing the YSZ powder to melt fully. The segmented APS coating had a typical lamellar structure similar to the traditional APS coatings. For the as-sprayed coating, the splat–splat interactions were not apparent and even disappeared, whereas the consecutive splats were well-bonded. This is probably because of the continuous growth of columnar crystals between the lamellas. Some columnar crystals even penetrated several lamellas and formed a good connection between the lamellas. There were also few microstructural defects such as cracks and pores persisting in the coating. From the picture, some horizontal cracks and vertical cracks were obviously detected, which were beneficial to improving the service life of the ceramic coat.

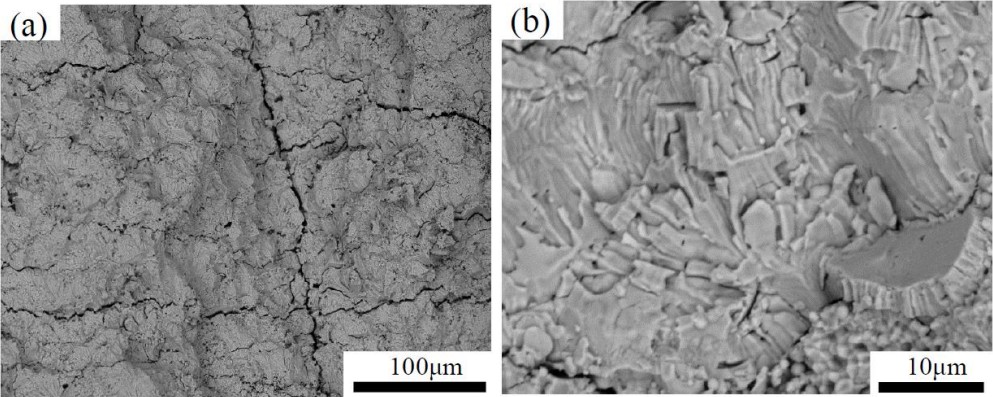

**Figure 5.** Fracture morphology of free-standing specimens: (**a**) at low (left) and (**b**) high (right) magnification).

Figure 6a–f shows the fracture morphologies of as-sprayed coatings and the samples after thermal aging for 24 h from 1200 to 1600 °C, respectively. As-sprayed coating consisted of columnar crystals, and the bridging boundaries between splats were in a close combination. Limited interfaces could be found. After thermal exposure to high temperatures, the columnar crystals diffused and grew rapidly. In addition, the growth rate of grains varied from fast to slow. Accompanied by a temperature increase from 1200 to 1500 °C, the grain sizes and morphologies changed conspicuously, and the columnar grain characteristics were no longer obvious. It was evident that porosity in the coating was significantly reduced and the compactness was improved greatly. Moreover, along with the aging temperature increasing to 1600 °C, there was a trend that the columnar crystals were slowly transformed into equiaxed crystals.

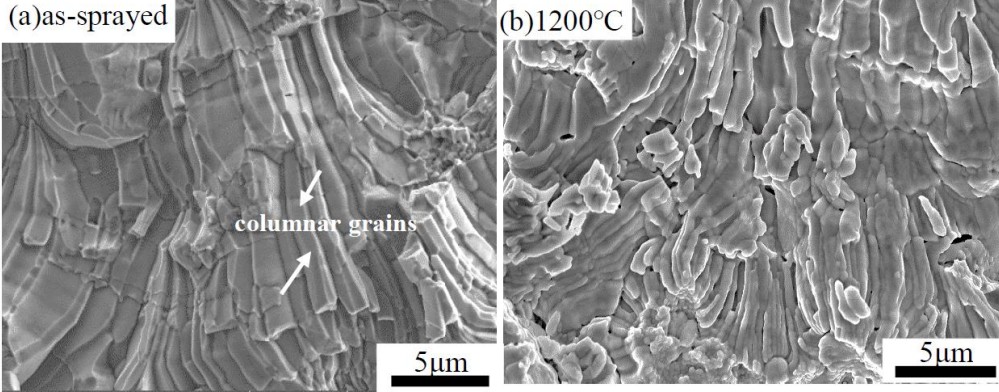

**Figure 6.** *Cont.*

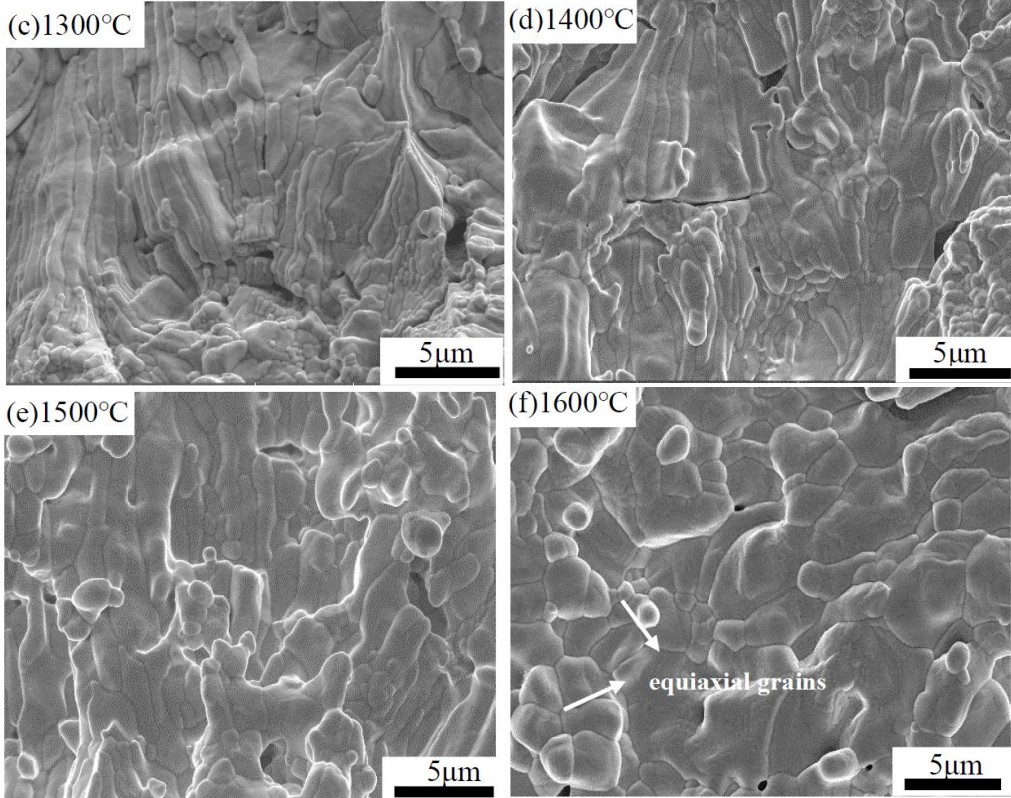

**Figure 6.** Fracture morphologies of YSZ top coatings before and after thermal aging at different temperatures. (**a**) as-sprayed; (**b**) at 1200 °C for 24 h; (**c**) at 1300 °C for 24 h; (**d**) at 1400 °C for 24 h; (**e**) at 1500 °C for 24 h; (**f**) at 1600 °C for 24.

Figure 7a–f shows the fracture morphologies of as-sprayed coatings and the samples after aging at 1550 °C from 20 h to 100 h, respectively. After 1550 °C heat treatment for 20 h, the grain size was much larger than that of the as-sprayed coating. Interlaminar cracks and intralaminar cracks healed apparently. The columnar crystals gradually transformed into equiaxed crystals and the number of equiaxed crystals increased with the thermal aging time rising. This significant change in microstructure indicated that the coating sintering was distinct. Grains grew obviously with increasing aging temperature and the crystal boundary was visible. However, for coatings treated at 1550 °C, grains grew quickly during the initial stage and slower after 20 h, where the grain morphology began to change. Together, they indicated that grain growth is more sensitive to thermally heated temperatures. In addition, after 1550 °C/40 h thermal treatment, the porosity of coatings increased and microcracks began to increase. As previously mentioned, the phase transformation from the t-YSZ to m-YSZ phase accompanied by volume expansion may lead to the nucleation and propagation of pores.

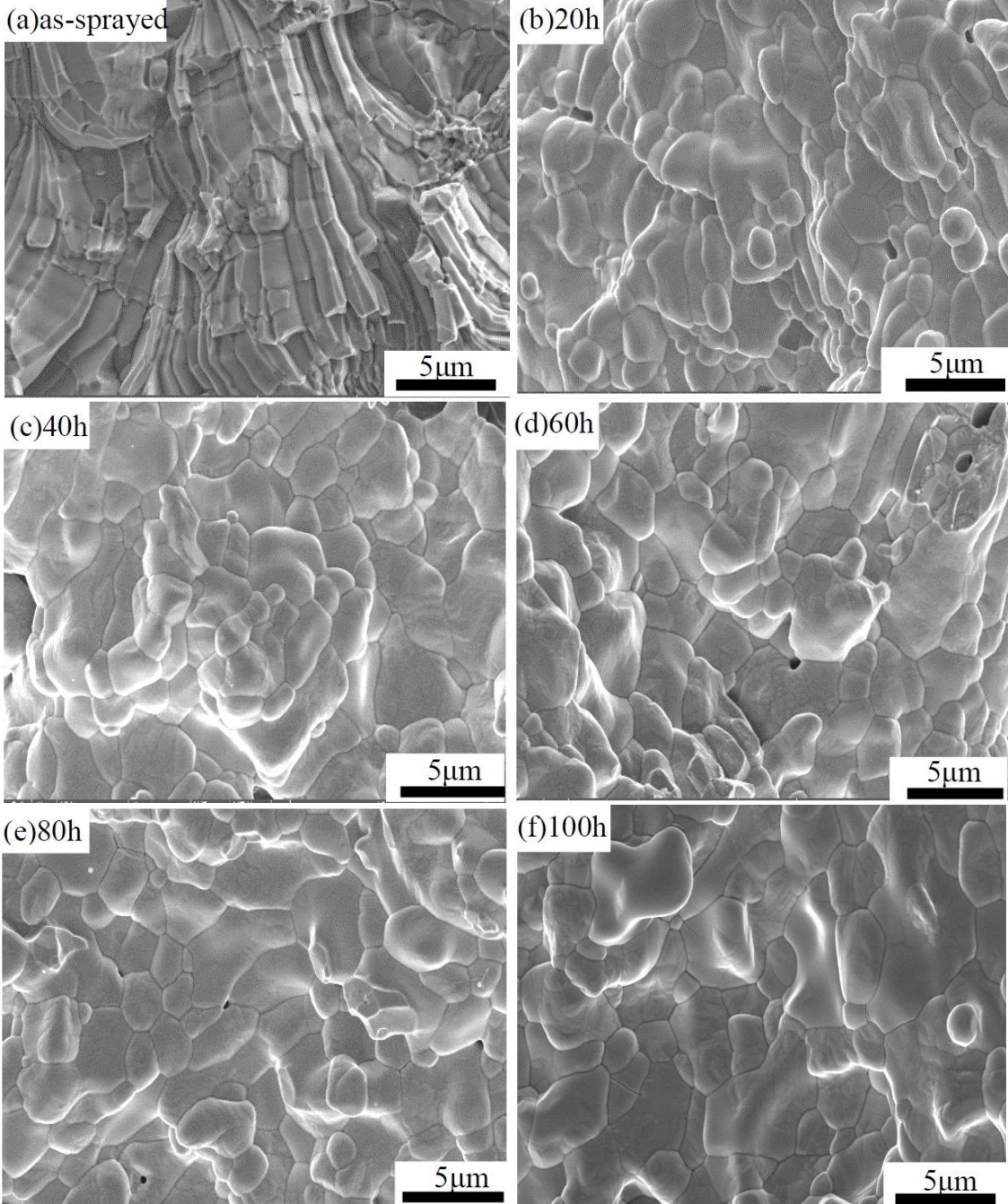

**Figure 7.** Fracture morphologies of YSZ top coatings before and after thermal aging for different times. (**a**) as-sprayed; (**b**) at 1550 °C for 20 h; (**c**) at 1550 °C for 40 h; (**d**) at 1550 °C for 60 h; (**e**) at 1550 °C for 80 h; (**f**) at 1550 °C for 100 h.

Dalach et al. [31] reported that the formation mechanism of micropores may be the aggregation of vacancies in grains during high-temperature thermal exposure. Vacancy accumulation led to lattice collapse and micropore formation, which caused a decrease of vacancy concentration in the crystals. At the same time, new vacancies were created to maintain the equilibrium concentration of vacancies in the crystal. Finally, the increase and aggregation of oxygen vacancies formed vacant clusters, which promoted the diffusion of atoms near the grain boundary. We can also observe some crystal cracking, which can be explained by the different thermal expansion coefficients in different crystallization directions, so that with improved temperature, the linear expansion effect of grains on both sides of

the grain boundary was different, resulting in stress at the grain boundary. The presence of greater stress eventually led to cracks at the grain boundary and final crystal cracking. It can be concluded that the grain size, grain distribution and coating porosity were influenced greatly by the thermal environment associated with the spreading of plasma-sprayed droplets into splat–splat and the latter's rapid solidification.

Figure 8 shows the cross-sectional microstructures of the as-sprayed coating and the samples after thermal exposure. The as-sprayed coating had large penetrating cracks and the measured crack gap was approximately up to 4 μm, in favor of releasing the thermal mismatch stress during heat treatment. As expected, when the thermal aging temperature increased from 1200 to 1600 °C, the porosity decreased and microcracks surrounding the penetration crack regions began to heal. When the samples were heat-treated at 1600 °C/24 h, the segmentation cracks' density dropped markedly, but for the coatings' thermal exposure at 1550 °C, the coating porosity first declined and then appeared. After 40 h thermal treatment, the coating gradually coarsened, causing microcracks and pores in the coating to appear again. This may be due to the fact that the coating not only underwent sintering, but also with prolonged aging time, the sample suffered phase transformation after heat treatment and formed a monoclinic phase. The phase transformation of t-YSZ into the m-YSZ phase was accompanied by volume expansion, resulting in the formation of more microcracks in the coating. This was consistent with the XRD test results.

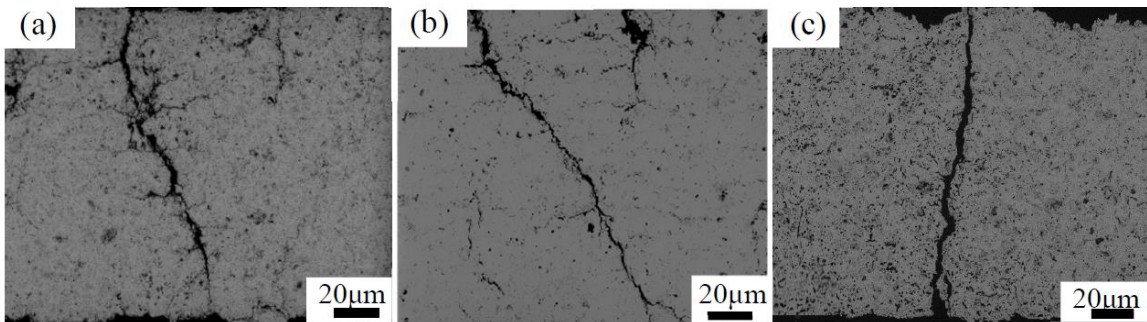

**Figure 8.** Microstructure of as-sprayed and thermally aged APS coatings: (**a**) as-sprayed; (**b**) at 1600 °C for 24 h; (**c**) at 1550 °C for 100 h.

To further investigate the thermal aging behavior of TBCs, Figures 9–11 show the SEM images, EBSD phase maps and orientation maps of the polished cross-sections of the as-sprayed coatings and the samples after thermal exposure. The EBSD images show the phase composition of each sample, content of each phase, the distribution of pores and size and features of the crystal grains. Different colors represent different phase compositions. The red areas indicate the tetragonal phase, blue areas indicate the monoclinic phase and black areas indicate cracks and pores. The as-sprayed coatings mainly consisted of a tetragonal phase, which was caused by the rapid cooling and solidification of inflight particles. A large number of dense columnar grains existed in the as-deposited YSZ coating with an average grain size of 1.22 μm. The grain sizes of the as-sprayed segmented coating were smaller than that of the traditional APS coating. As previous studies have shown, small grains are beneficial to the inhibition of phase transition [9]. This resistance could play a dominant role in the kinetics of the observed increase in the monoclinic content with time and temperature as the transformation becomes much more favorable after sintering grain growth. Seen from Figure 10, the coating began to undergo phase transformation after 24 h heat treatment at 1600 °C. Meanwhile, few pores were observed. For the samples that were thermally aged at 1550 °C/100 h, a mixture of columnar grains and equiaxed grains was detected. It can be seen that a greater number of pores appeared in the coating due to the phase transition. The calculated grain sizes of the two distinct monoclinic and tetragonal phases were approximately 0.72 and 1.63 μm, respectively. The percentage of the m-YSZ phase was approximately 12.4%. Witz et al. [22] suggested that the phase decomposition comes together with an

increase of its microstrain, with values high enough to initiate cracks that, after propagation, leave coating failure. The images also show that there was no preferred grain orientation, whether or not the coating was thermally treated. The content of the monoclinic phase, tetragonal phase and grain size can be obtained from the data calculated by EBSD, all shown in Table 2.

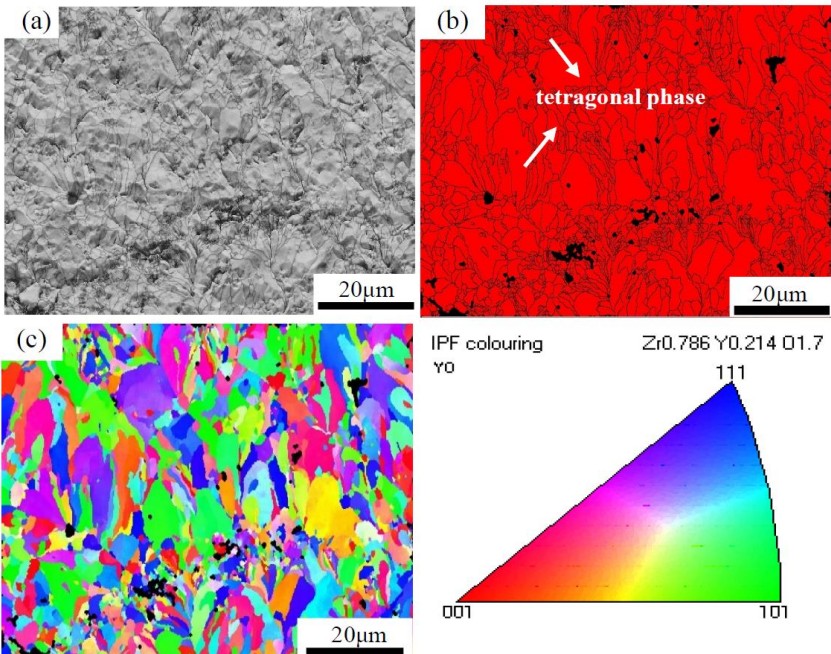

**Figure 9.** (**a**) SEM images, (**b**) EBSD phase maps of cross-sections of the as-sprayed coating and (**c**) orientation maps, Y-direction. (Tetragonal phase is meant as a group of t, t′ and t″ phases).

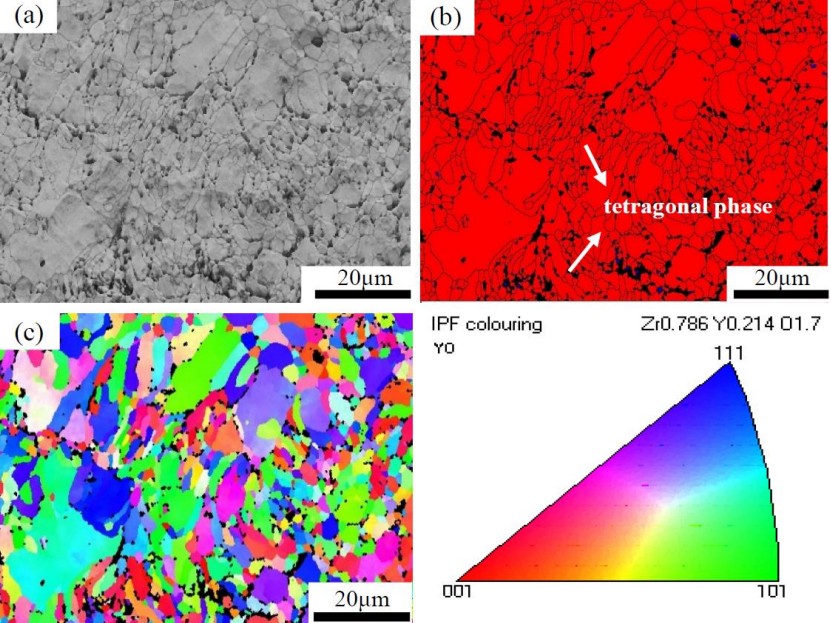

**Figure 10.** (**a**) SEM images, (**b**) EBSD phase maps of cross-sections of the top coating after 1600 °C/24 h thermal exposure and (**c**) orientation maps, Y-direction. (Tetragonal phase is meant as a group of t, t′ and t″ phases).

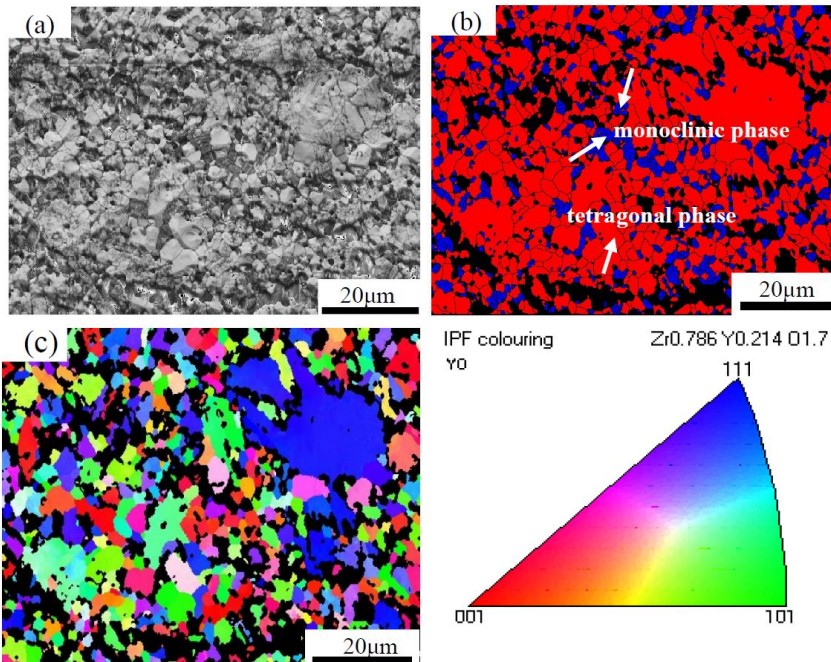

**Figure 11.** (**a**) SEM images, (**b**) EBSD phase maps of cross-sections of the top coating after 1550 °C/100 h thermal exposure and (**c**) orientation maps, Y-direction. (Tetragonal phase is meant as a group of t, t′ and t″ phases).

**Table 2.** The microstructure parameters of coatings before and after thermal treatment.

| Coating | Monoclinic Phase (%) | Tetragonal Phase (%) | Grain Size (μm) |
|---|---|---|---|
| As-sprayed | 0 | 98.1 | T: 1.22 |
| 1600 °C/24 h | 0.15 | 92.5 | T: 1.71<br>m: 0.55<br>T + m: 1.67 |
| 1550 °C/100 h | 12.4 | 63.8 | T: 1.63<br>m: 0.72<br>T + m: 1.05 |

Note: T (tetragonal phase) is meant as a group of t, t′ and t″ phases; m (monoclinic phase); T + m represents all grains in the coating.

Sintering resulted in the decrease of coating porosity, volume shrinkage, grain size increment and increase of density and strength of coatings. Grain growth was not the mutual bond of small grains, but the result of grain boundary movement. The grain growth process was accompanied by the shrinkage or disappearance of another part of the grain. When small grains grew into large grains, causing a decrease in the grain boundary area, the interface free energy decreased, and the grain size increased. The core of grain growth is the increase in grain average size. From Figures 6, 7 and 11, it can be seen that the individual grain size increased abnormally, potentially due to the secondary recrystallization of the sintering process. When the normal grain growth stopped, individual large grains with multilateral boundaries grew abnormally by absorbing small grains, pores and impurities. In order to analyze the grain growth behavior accurately, quantitative data on the grain size corresponding to the tetragonal phase (T), monoclinic phase (m), T + m-phase are given statistically in Figure 12. The phase composition of the as-sprayed coating was single, but after heat treatment at 1550 °C for 100 h, it was mainly composed of monoclinic and tetragonal phases. As mentioned earlier, the percentage of m-YSZ was approximately 12.4%. For these crystalline grains with an m-phase inside, the grain size was concentrated in 0.5–0.7 μm. It is worth noting that phase transition was more likely to take place at the crystal boundary. This was because of the irregular arrangement of atoms on the



grain boundaries, leaving many defects such as vacancies, bit band and bond deformation, which left it in a state of stress distortion [32]. Therefore, the higher energy order made the grain boundary become the preferred nucleation region in the phase transition.

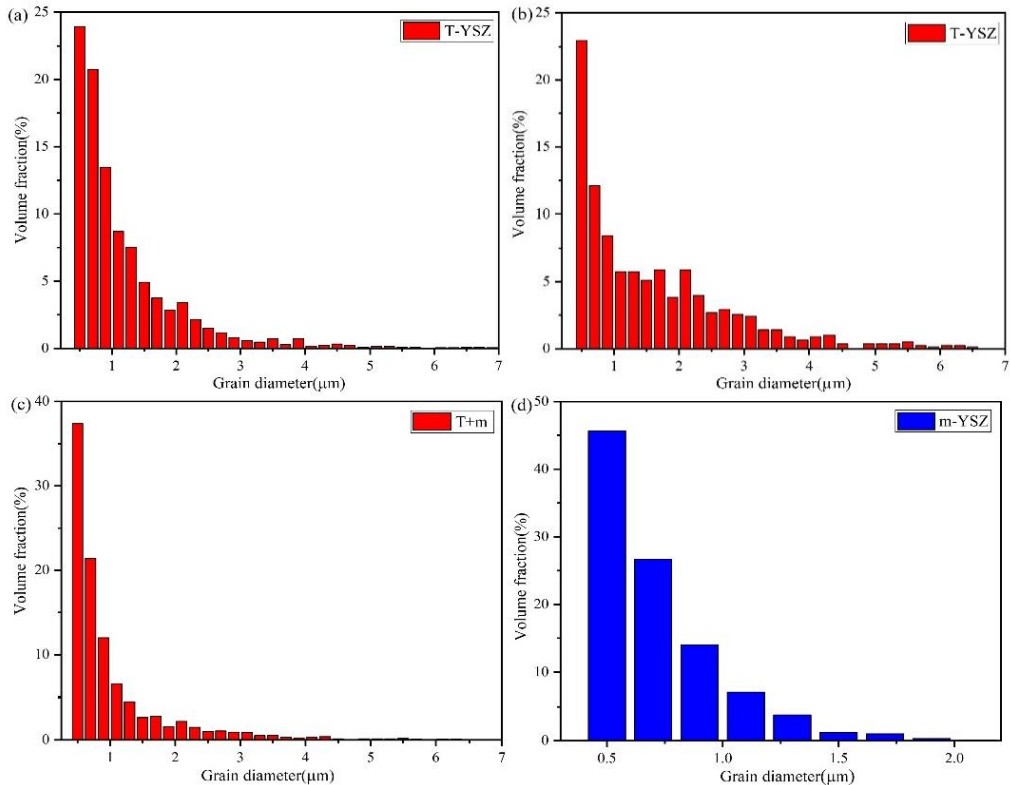

**Figure 12.** Grain size of YSZ coatings before and after thermal treatment: (**a**) as-sprayed; (**b–d**) 1550 °C/100 h. T (tetragonal phase) is meant as a group of t, t′ and t″ phases; m (monoclinic phase); T + m represents all grains in the coating.

### 3.3. Mechanical Properties

A larger number of studies have shown that after the high-temperature treatment, the phase composition, pore structures, crack states and grain morphology of the ceramic coating changed significantly, which had an important influence on the mechanical properties of the YSZ coating. Figure 13 showed the relationship of mechanical properties (Vickers hardness) of APS coating with heat treatment temperature and time. The hardness of the coating region composed of molten particles was obviously higher than that of semimolten and nonmolten particles. It can be seen that before heat treatment the mean Vickers hardness value of as-sprayed APS coating is 948 ± 37 HV0.3, it was similar to that of the SPS coating but slightly higher than that of traditional APS coating fabricated with F4MB-XL torch (919 ± 21 HV0.3) [11]. This may be due to the fact that the Triplex Pro™-200 torch provided higher power to make the coating composed of finer grains and preferable melting state of powder particles during spraying, which facilitated the coating having a higher Vickers hardness.

Seen from Figure 13a, there was a significant increase in hardness value with the constant increase in thermal aging temperature. After 24 h thermal exposure at 1300 °C, the top coating exhibited the highest mean hardness of approximately 1225 ± 58 HV0.3. This may have a close relationship with the coating sintering during heat treatment. Thermally treated samples became denser, and the augment of coating hardness came at the cost of increasing the thermal conductivity [33]. Continuing to increase the temperature, the value dropped slightly and fluctuation was small. Figure 13b shows the Vickers hardness value of the APS coating after thermal aging at 1550 °C for different times. The Vickers hardness of coating increased rapidly at first and reached the maximum after 20 h of heat treatment

(1246 ± 36 HV0.3). Compared to the as-deposited coating, it increased by 31%. Then, it decreased rapidly with the prolongation of heat treatment time, remaining relatively stable at approximately 1091 ± 23 HV0.3. Briefly, the improvement of hardness was attributed to the coating sintering along with the elevated splat–splat bonding and healing of defects like pores and microcracks. However, for the thermally treated coatings with prolonged aging time, the mechanical properties significantly decreased due to the formation of the m-YSZ phase. The phase transformation resulted in irreversible volume expansion, which led to the appearance and propagation of microcracks in the coating, causing a decrease in coating hardness. This was consistent with previous experimental results. It can be concluded that the grain growth of coatings was more sensitive to thermal aging temperature and sintering was more likely.

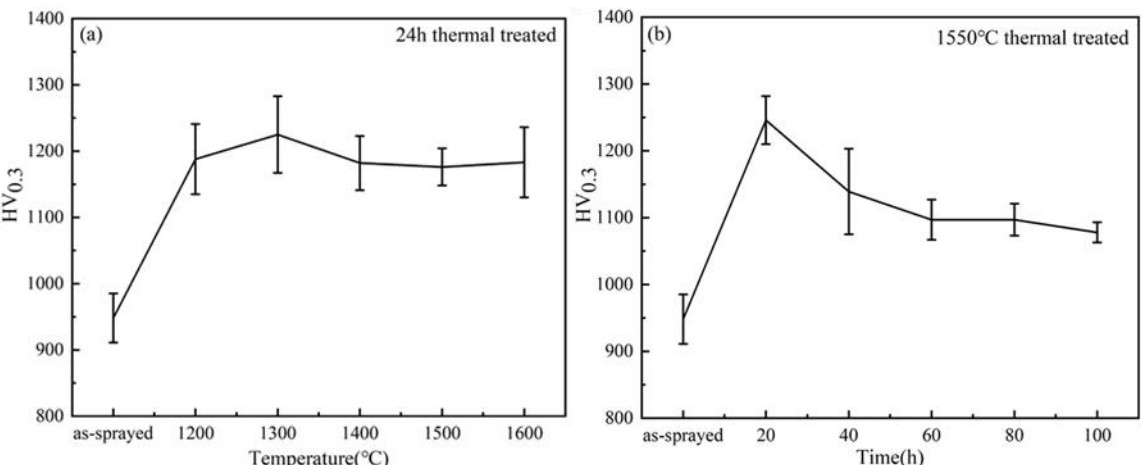

**Figure 13.** Variation of Vickers hardness of APS YSZ coatings with thermal aging treatment: (**a**) samples after thermal aging treated at different temperatures for 24 h, and (**b**) thermal aging treated at 1550 °C for different times.

## 4. Conclusions

Segmented-crack YSZ TBC$_S$ were deposited by a specific APS process using the TriplexPro-200 torch. This paper presented the results of phase composition, microstructure evolution and mechanical properties before and after thermal treatment. The influence of thermal aging time and temperature was investigated. The main conclusions were drawn as follows:

(1)  The segmented APS YSZ coatings exhibited similar high-temperature phase stability to the conventional APS coating. During thermal exposure, the initial metastable tetragonal (t′-YSZ) phase of the coating gradually partitioned into equilibrium tetragonal (t-YSZ) and cubic (c-YSZ) due to yttrium diffusion, and as the temperature improved, the c-YSZ remained or transformed into another yttrium-rich tetragonal (t″-YSZ) phase. The transformation of t-YSZ to monoclinic phase (m-YSZ) occurred after 1550 °C/40 h heat treatment, and the content of the m-YSZ phase increased with the prolongation of the thermal exposure time.

(2)  During thermal exposure, cracks and pores proceeded along the grain boundaries, especially surrounding the small grains. In service, t-YSZ to m-YSZ phase transition easily occurred at the grain boundary because the atomic arrangement on the grain boundary is looser than in the grain.

(3)  The segmented APS YSZ coatings had good bonding between the lamellas, which caused the mean Vickers hardness value was higher than that of conventional APS YSZ coatings. The Vickers hardness increased apparently after thermal aging treatment due to the sintering of the coating. However, a large number of m-YSZ phases formed with the extension of thermal aging time, and the transformation of t-YSZ into m-YSZ produced irreversible volume expansion, which induced the initiation and propagation of microcracks and resulted in the decrease of the mechanical properties of the coatings.

**Author Contributions:** Conceptualization, S.T. (Shiqian Tao) and S.T. (Shunyan Tao); data curation, S.T. (Shunyan Tao), X.Z., W.L., F.S., H.Z., Y.Z., and J.N.; formal analysis, S.T. (Shiqian Tao) and J.Y.; writing—original draft preparation, S.T. (Shiqian Tao); writing—review and editing, J.Y., S.T. (Shunyan Tao), and K.Y. All authors have read and agreed to the published version of the manuscript.

**Funding:** This research was funded by National Science and Technology Major Project (No. 2017-VI-0010-0082), National Natural Science Foundation of China (NSFC) (No. 51701235 and No. 51971148), and Science and Technology Innovation of Shanghai (No. 18511108702).

**Acknowledgments:** The authors are grateful for the technical assistance provided by Shiqian Tao.

**Conflicts of Interest:** The authors declare no conflict of interest.

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
