# Peer review of "Thermal Stability of Plasma-Sprayed Thick Thermal Barrier Coatings Using Triplex ProTM-200 Torch"

_coatings, doi:10.3390/coatings10090894_

Round 1
Reviewer 1 Report
This paper has the potential to highlight differences in thermal stability of low-porosity segmented APS TBCs (using the triplex torch) compared with conventional APS YSZ TBCs. Unfortunately, this paper presents results of phase stability and sintering without distinguishing how these results are different from conventional APS TBCs. In particular, no evidence is provided that the observed phase stability is any different from that observed in conventional APS YSZ TBCs, which has been reported many times in the literature. While the more obviously different fracture behavior and sintering grain growth could have been a focus, the conclusions all are based on phase stability results that appear to be no different than those observed for conventional APS TBCs. In addition, the paper also does not reconcile its reporting of only two phases, metastable tetragonal (t’) and monoclinic (m) when the vast literature on this subject indicates that t’ does not directly transform to m, but first partitions into a Y-deficient tetragonal (t) and a Y-rich cubic (c) after which the transformable t can transform into m. These are significant issues that must be addressed before this paper is acceptable for publication and are discussed in more detail below:
Major Issues
- The subject of the paper is the thermal stability of the low-porosity, segmented APS YSZ TBCs produced by the triplexpro torch; however, no indication is given in the paper that discriminates the thermal stability of these coatings with conventional APS YSZ TBCs. This is a significant issue because there are many published papers on the phase stability of YSZ in many forms, including APS coatings. Therefore, this paper would merit publication only by establishing differences in thermal stability compared with conventional coatings. Unfortunately, the paper provides no evidence that the YSZ phase stability is any different than that observed and reported many times for conventional YSZ coatings. The authors could certainly make a case for differences in sintering grain growth and fracture behavior, yet the conclusions center on the phase stability that appears to show no different behavior than that reported many times for conventional APS YSZ TBCs. Therefore, it is essential that the authors either clearly establish important differences in phase stability for their coatings or switch the focus of the discussion and conclusions to the fracture and sintering behavior, for which they could make a good case for different behavior.
- The authors make an oversimplification of the phase transformations that occur upon YSZ heat treatments. They only consider two phases, a transition from t’ to m without considering other phases and the important roles those phases play in phase stability. It is well established that t’ does not directly transform to m. The t’ is a metastable phase commonly referred to as non-transformable tetragonal. The t’ phase is desirable specifically because it does not transform directly to m. Many papers have shown that upon high temperature treatment the t’ phase partitions between a Y-poor t phase and a Y-rich cubic phase that occurs with Y diffusion. Unlike the t’ phase, the equilibrium t phase is directly transformable to the monoclinic phase, although that transformation is inhibited for small particle/grain sizes. The authors seem to initially refer to these transformations correctly in the introduction (lines 77-80) but then ignores the correct transformation kinetics in the rest of the paper. While the t and c phases do not show striking changes in the x-ray diffraction patterns, they are revealed by close inspection, especially in the 30 and 74 degree regions of the diffraction pattern. One example (among many) of identifying the t and c phases in the diffraction pattern is in the paper by G. Witz et al., “Phase Evolution in Yttria-Stabilized Zirconia Thermal Barrier Coatings Studied by Rietveld Refinement of X-Ray Powder Diffraction Patterns,” J. Am. Ceram. Soc., 90, 2935-40 (2007). Diffraction peaks for the t and c phases may be present in the Fig. 2a of the paper being reviewed, but it is not displayed large (or clearly) enough to inspect. In any event, there is no evidence that the metastable t’ phase would be retained at all for the long duration 1550 and 1600C heat treatments, yet the authors still claim majority t’ phase. It would also be very useful if the authors included energy dispersive x-ray spectroscopy (EDS) results along with their SEM results as these would clearly distinguish the tetragonal and cubic phases (by their Y-deficient and Y-rich content) and would show the relationship between grain size and phase.
- The authors should consider the well known resistance to tetragonal to monoclinic transformation by small grain/particle sizes. This resistance could play a dominant role in the kinetics of their observed increase in monoclinic content with time and temperature as the transformation will become much more favorable after sintering grain growth.
- The authors need to make a case for studying and emphasizing phase transformations up to 1600C. It is well known that the t’ phase transformations described above occur at temperatures above 1300C. It is specifically for this reason that YSZ coatings are not considered a viable option for performance above 1300C. For higher temperature applications, alternate compositions with better high temperature stability, e.g., zirconates and hexaaluminates, are preferred. Therefore, the authors should answer the question as to why they are studying the phase stability at temperatures so much higher than any applications that would be considered for YSZ.
Minor Issues
- The authors should mention there are tradeoffs when you increase the TBC density. The coating hardness will increase, but at a cost of increasing the thermal conductivity.
- Similarly, there are tradeoffs with the branching cracks: they will increase compliance, but will make the coating susceptible to erosion and foreign object damage.
- Line 129: “Instroin” should be “Instron”.
- 2a inset needs to be sharper and with larger font axis labeling.
- Lines 176-177 are confusing. First it is stated that splat boundaries are visible, then next sentence states they are not visible. The comparison is unclear.
- The meaning of an absence of a segmentation crack in Fig. 7b is unclear. Does this indicate that there was a crack but that it healed? The text suggests that branching cracks disappeared, so it might be more illustrative to show a partially healed segmentation crack with completely healed branching cracks.
- At least one of the micrographs in Fig. 6 (perhaps Fig. 6f) should label representative grains as t, c, and m phase. [There should not be t’ at this point.] Similarly, the correct phases should be identified in Table 2.
Reviewer 2 Report
- Conflicting analysis is made on mechanical properties. Authors seem to make little vague analysis and explanation with results. As thermal aging temperature is increased to 1600C there is no change in Hd (around 1200Hv) that is little puzzling as higher temperature is supposed to have more microstructural changes and sintering effects. Even the grain growth is stated to be absent which is a bit puzzling.
- Authors claim that the grain growth effect is more sensitive to temperatures, but their results do not support their views.
- However, for 1550C aging, authors try to explain the hardness variation with phase transformation, microcrack effects and also grain growth effect, not sure what is the actual reason. Should investigate more and pinpoint the actual issue.
- Microcrack formation and porosity change have been used to explain the experimental observations at many places in the paper. However, no quantitative results for the two are included to convince the readers as to why and how these two varied from samples to samples and due to various thermal aging effects. A better and convincing explanation would have emerged the. No images is shown for these microstructural changes due to thermal aging consequences.
- Another important points the authors must note and explain. How the fracture surfaces are examined? How did they get to see fracture morphology? Are they confusing the fracture surface with microstructural images? This needs to be clear. When a broken/fractured part of a sample is examined un SEM, this is termed as fracture surface examinations?? what are they refereeing to??
- Table 2, the sum of m and t phases are well below 1 and authors should explain the reasons while the t phase in As sprayed structure is close to 1?
- Line 267 &268, from which observations/results, the authors find rising density, rising strength, porosity dcrease? volume decrease? Authors should quantify these claims, rather speaking loosely
Reviewer 3 Report
Authors have made an attempt to understand the sintering behaviour of thick plasma sprayed segmented thermal barrier coatings. The topic of sintering within TBCs is of high interest. Despite of this, the paper has several shortcomings and therefore should not be published in coatings journal in its current form.
My main objections are as follows:
- The introduction lacks a clear motivation to do this study. For example, why did authors select such a high temperature to perform these experiments is not motivated. This should be explained by clearly giving examples of real-life conditions that TBCs experience. Moreover, the latest reference cited in the article is from 2017. Does this mean there are no studies on TBCs during the last 3-4 years addressing similar research question?
- The experimental section is very poorly described. No details are provided on how the experiments are conducted and especially how the samples are prepared for each characterization test. For examples, preparing TBC sample for EBSD is difficult due to the excessive charging effect on ceramic coating and therefore it would be interest for readers to know how authors managed to obtain such neat EBSD images/maps. Also, how did authors measure the porosity and segmentation density is not clear.
- Results and discussion section only include ONLY data interpretation and no detailed scientific discussion or mechanisms are described.
- Figures are not labelled at all and it is very hard to locate different features on figures that authors are talking about in the text.
- Authors have shown very interesting results on transformation of columnar grains to equiaxed with increased time and temperature. However, authors did not explain the detailed mechanism behind such behaviour. Furthermore, it would also be of interest to readers to know if such a behaviour is also observed in literature and appropriate references to convey this is also missing.
Round 2
Reviewer 1 Report
The authors have made significant improvements in their manuscript, in particular, a clearer, more accurate description of the YSZ phase transitions and some useful clarifying figure modifications (particularly the higher resolution diffraction patterns in Fig. 3). For the most part, the authors gave good answers in response to this reviewer’s questions. However, there remains ample lack of clarity in the descriptions of the YSZ phase transitions and a needed refocus of the conclusions was not performed. In addition, many of the good clarifications by the authors in their responses were not incorporated into the manuscript. In most of those cases, incorporating some of the wording of their responses into the text would sufficiently address those issues. While significant improvements in the manuscript were made, these issues (described in more detail below) need addressing for the manuscript to be acceptable for publication.
Significant Remaining Issues
- To a large extent, the authors have made the changes needed for a more accurate description of the YSZ phase transformations with the additional material at the bottom of p. 5, and the manuscript no longer suggests incorrectly that the primary transformation is directly from t’ to m. However, there remain significant portions of the manuscript where the original inaccurate description has not yet been changed and where the new description remains confusing. The sections that still need revision clarification include:
- Fig. 2 needs a more specific identification of the tetragonal phase peaks in the diffraction patterns because tetragonal is generally understood to be the transformable t phase, whereas these patterns contain a mixture of t, t’, and t’’ phases. Therefore, the peaks should be labeled by their specific phase or else the caption should note that tetragonal is meant as a grouping of all three phases. For example, the peaks are primarily t’ for the as-deposited coating, whereas the peaks are primarily a mixture of t and t’’ (but not t’) for the longest annealing conditions.
- The listing of the phases in the last column of Table 2 is confusing. First of all, lower case letters t and m should be used to be consistent with the text (not uppercase T and M). Secondly, there are two entries for T+M in the last row when it is not clear what the meaning of the T+M phase is (and why it has two entries with different grain sizes. In addition, this is another case where there seems to be no discrimination between t, t’, and t’’. This produces an inaccurate characterization of the phases because typically the large grains are c (or t’’), not t as indicated in the table, while small grains are m or t (since m is formed by transformation of t). As written, Table 2 is impossible to understand.
- A similar problem occurs with Figures 9-11 along with the supporting discussion. First of all the figures should have a legend (or mention in the caption) showing that red indicates tetragonal and blue indicates monoclinic. Secondly the captions are mislabeled because each figure b is a phase map while each figure c is an orientation map while the captions state the opposite. Most importantly, the authors once again need to indicate what specific phase(s) they mean by tetragonal: t, t’, or t’’ or all of them. Just labeling as tetragonal is not helpful to the reader. This is particularly important in establishing a correlation between grain phase and grain size. Research by others have shown that t’ partitions into smaller t grains and larger c grains. Because t transforms to monoclinic and c to t’’, the m and t grains will be small while the c and t’’ grains will be large. Therefore it is essential to separate the labeling of the tetragonal phases (and also to make some sense of the M+T designation).
- The same problem occurs with Fig. 12. First of all the legend should indicate YSZ, not ZrO2. Most important is the recurring problem of a nonspecific assignment of t phase. Fig. 12a, as-coated, should be t’. There should be little t’ in the other figures which should contain small grain t (that can transform into small grain m) and large grain t’’ (that transformed from a large grain c precursor). A better description of t+m is again needed.
- The same clarifications to the phase – grain size correlations are needed in the text in lines 340-346.
- Some of the clarifications in the authors’ response should be incorporated into the text so that these issues are clear to the reader as well as this reviewer:
- From response 4, the authors should include in the text a rewording of their good response to why they made measurements at so much higher temperatures than the actual use temperature of YSZ: “Although the t’-ZrO2 phase has begun to decompose at around 1300℃, it's easier to accelerate the phase transition at higher temperature.”
- Most importantly, the authors should incorporate aspects of their response 1 where they stress that while they do not observe different or new phase stability behavior in their segmented coatings, they do observe significant differences in microstructure and mechanical properties (hardness) after heat treatments. Unfortunately, the conclusions were not changed accordingly and still emphasize the previously known phase behavior. Therefore, the conclusions need to be rewritten to reflect an emphasis on differences (and similarities) observed in the segmented coatings compared to conventional APS coatings. The first conclusion therefore needs to be changed from a summary of phase behavior to a statement that the phase behavior was no different than observed in other YSZ coatings. That should than be followed by conclusions that different microstructure and mechanical property behavior with heat treatment were observed for the segmented coatings with a brief summary of what those differences are.
Minor Issues
- In line 54, the word “wildly” should be removed (wildly means recklessly). Perhaps the authors meant widely – in any event, deleting is the best option.
- In line 171, “yttrium-sufficient” should be “yttrium-rich”.
Reviewer 3 Report
Thanks for taknig care of the comments. The paper can be published in its current form.
Author Response
Thank you very much spending your precious time to complete the review.
Round 3
Reviewer 1 Report
The authors have fully addressed the issues raised by this reviewer. I believe the clarifications will be of significant benefit to the readers and I commend the authors on their efforts that should result in a clearer, more focused paper.